# Potential of Cell-Penetrating Peptide-Conjugated Antisense Oligonucleotides for the Treatment of SMA

**DOI:** 10.3390/molecules29112658

**Published:** 2024-06-04

**Authors:** Jamie Leckie, Toshifumi Yokota

**Affiliations:** 1Department of Medical Genetics, Faculty of Medicine and Dentistry, University of Alberta, Edmonton, AB T6G 2H7, Canada; 2The Friends of Garrett Cumming Research & Muscular Dystrophy Canada HM Toupin Neurological Sciences Research, Edmonton, AB T6G 2H7, Canada

**Keywords:** cell-penetrating peptides (CPPs), antisense oligonucleotides (ASOs), spinal muscular atrophy (SMA), delivery, phosphorodiamidate morpholino oligomers (PMOs), DG9

## Abstract

Spinal muscular atrophy (SMA) is a severe neuromuscular disorder that is caused by mutations in the survival motor neuron 1 (*SMN1*) gene, hindering the production of functional survival motor neuron (SMN) proteins. Antisense oligonucleotides (ASOs), a versatile DNA-like drug, are adept at binding to target RNA to prevent translation or promote alternative splicing. Nusinersen is an FDA-approved ASO for the treatment of SMA. It effectively promotes alternative splicing in pre-mRNA transcribed from the *SMN2* gene, an analog of the *SMN1* gene, to produce a greater amount of full-length SMN protein, to compensate for the loss of functional protein translated from *SMN1*. Despite its efficacy in ameliorating SMA symptoms, the cellular uptake of these ASOs is suboptimal, and their inability to penetrate the CNS necessitates invasive lumbar punctures. Cell-penetrating peptides (CPPs), which can be conjugated to ASOs, represent a promising approach to improve the efficiency of these treatments for SMA and have the potential to transverse the blood–brain barrier to circumvent the need for intrusive intrathecal injections and their associated adverse effects. This review provides a comprehensive analysis of ASO therapies, their application for the treatment of SMA, and the encouraging potential of CPPs as delivery systems to improve ASO uptake and overall efficiency.

## 1. Introduction

Spinal muscular atrophy (SMA) is a rare genetic condition that is characterized by progressive muscle weakness and respiratory complications that often lead to death [1]. The SMA phenotype is the result of a significant deficiency in full-length survival motor neuron (SMN) protein. Nusinersen, an ASO designed to address the present deficiency by improving the splicing efficiency of an SMN gene analog that typically is translated into a nonfunctional SMN protein, was FDA-approved in 2016 [2]. While demonstrating efficacy in improving the SMA phenotype in treated patients, Nusinersen must be administered intrathecally for CNS delivery, accompanied by notable side effects [3]. Moreover, the inherent poor cellular uptake of ASOs limits their efficacy at lower doses [4].

To overcome these challenges, several delivery systems have been explored to enhance ASO cellular uptake for improved efficiency. Among these, cell-penetrating peptides (CPPs) emerged as a promising small-molecule carrier. When directly conjugated to neutrally charged ASOs, CPPs were seen to not only improve ASO cellular uptake but also facilitate delivery into the CNS following system injection in SMA mice [5,6,7,8,9]. Despite the evident potential of CPPs to enhance ASO delivery, they have yet to be FDA-approved as a carrier due to concerns regarding their toxicity and immunogenicity [10]. This comprehensive review provides an overview of ASOs for the treatment of SMA and the observed potential of CPPs to improve their efficiency and avoid the need for invasive procedures.

## 2. Background on Spinal Muscular Atrophy 

SMA is an autosomal recessive neuromuscular disorder that is caused by the degeneration of spinal anterior horn cells [11]. The associated denervation of the limb, trunk, bulbar, and respiratory muscles lead to progressive weakness, difficulty swallowing, and respiratory problems [12]. The estimated prevalence of SMA is between 1 and 10 in 100,000 live births, with observed intercountry variability, and it is currently one of the leading inherited causes of infant death [13,14,15]. The SMA phenotype was discovered to be due to biallelic loss or biallelic pathogenic mutations in the *SMN1* gene [13,16]. The SMN protein encoded by the survival motor neuron 1 (*SMN1*) gene is evolutionarily conserved and ubiquitously expressed [17]. This 38 kDa protein is composed of a basic/lysine-rich domain, a tudor domain, a proline-rich domain, and a YG box. All four domains and the protein’s overall structure appear to be critical for the function of the SMN protein as pathogenic mutations causing SMA have been identified in each domain [18,19]. SMN is a multifunctional protein that plays a role in various cellular processes, including RNA metabolism [20,21,22], DNA recombination and repair [23,24], signal transduction [25,26], and intracellular trafficking [19,27,28]. 

The age of onset and severity of symptoms can vary greatly between patients diagnosed with SMA, primarily due to the compensatory capacity of alternative genes in mitigating the loss of functional proteins translated from *SMN1*. SMN is not only essential for motor neuron survival, as the absence of any SMN protein is embryonic lethal [29]. Fortunately, *SMN1* is not the only gene that encodes the SMN protein on the human genome. The *SMN2* gene is nearly identical to the *SMN1* gene. However, *SMN2* contains a single nucleotide alteration in exon 7 that disrupts the exon’s splice enhancer [30]. Due to this variation, the primary mRNA produced by *SMN2* lacks exon 7 and is translated into a truncated and nonfunctional protein [31]. Although exon skipping is favoured, small amounts of full-length functional SMN proteins translated from *SMN2* can compensate for the loss of functional protein from *SMN1* in SMA. Individuals can have varying copy numbers (between 0 and 8) of *SMN2* that impact its ability to compensate for loss of *SMN1*, with larger copy numbers of *SMN2* typically resulting in reduced disease severity [32,33]. Mutations in exon 7 of *SMN2*, which result in the creation of a novel splicing enhancer, can also influence the amount of full-length SMN proteins translated from *SMN2* transcripts [34]. Although dysfunctional SMN proteins transcribed from *SMN1* cause SMA, the severity of the disease is strongly associated with the individual’s *SMN2* copy number and sequence.

SMA has been classified into five types based on the age of symptom onset, disease severity, milestones achieved, and lifespan. Patients diagnosed with SMA type 1 begin to present with symptoms in the first 6 months of their lives and never achieve the ability to sit up on their own [35]. SMA type 1 is the most frequently diagnosed type and is also the leading cause of infantile death from a genetic disease [36]. Patients diagnosed with type 2 SMA begin to present with symptoms after 6 to 18 months of age. These patients experience progressive weakness in the arms and legs, progressive scoliosis, and restrictive lung disease due to weakening intercostal muscles. However, they have the ability to sit up on their own [35]. SMA types 3 and 4 are not associated with a reduced life expectancy, and although these patients experience progressive muscle weakness and atrophy, they achieve the ability to walk unaided [37]. Type 0 SMA, also known as prenatal SMA, is the most severe form of the disease. Prenatal SMA is associated with reduced fetal movements during pregnancy and severe muscle weakness and respiratory distress at birth [38,39]. In all prenatal SMA cases, patients die soon after birth. 

## 3. Antisense Oligonucleotides 

Short fragments of nucleic acid polymers, known as oligonucleotides, have become an incredibly useful tool for understanding gene function and represent a promising avenue for genetic disease therapeutics. ASOs are single-stranded DNA-like sequences, normally ranging from 12 to 25 nucleotides in length, that are designed to be complementary to the intended target sequence. With the proper design, these ASOs can specifically bind to their target sequence through Watson–Crick pairing, with minimal off-target pairing. The chemistry of the ASOs and their target binding site on RNA regulates their mechanism of action.

Once bound to their target RNA, ASOs can alter gene expression through two primary mechanisms, RNase H-mediated degradation of the target mRNA or steric blockage of specific sites on pre-mRNA or mRNA to modulate splicing or prevent translation (Figure 1). RNase H is a ubiquitous enzyme in all eukaryotic cells that is responsible for hydrolyzing RNA when it is bound to DNA during DNA replication [40]. ASOs can take advantage of RNase’s ability to degrade DNA by possessing a complementary sequence to that of the RNA intended to be degraded. Once the ASO-RNA heteroduplex is formed, RNase H can bind and hydrolyze the RNA so that it is unable to be translated into a disease-causing protein [41]. However, when all DNA bases in an ASO are chemically modified, RNase H is no longer able to recognize the DNA-RNA heteroduplex and elicit its enzymatic activity. In the absence of RNase H activity, ASOs bound to target RNA can block machinery involved in the translation or splicing of the RNA transcript [42,43]. Although steric blockage of translation machinery with ASOs has been observed to effectively reduce target protein translation [44], ASOs utilizing RNase H-mediated degradation are currently the preferred approach when a decrease in protein production is desired. ASOs have even been capable of increasing target protein expression by binding to 3′ and 5′ untranslated regions of pre-mRNA, resulting in improved mRNA stability [45].

Alternative splicing regulates which exons will be included in the mature mRNA product after processing. It is an essential biological process that allows for the translation of related but different proteins, with differing properties and functions, from the same DNA sequence [46]. Antisense oligonucleotides can promote specific splicing events by targeting splice sites and splicing enhancers and silencers [47]. ASOs can correct reading frames that are disrupted by insertions or deletions by promoting the exclusion of the disrupted exon, resulting in the translation of a truncated, yet functional, protein. Multiple ASOs that restore the reading frame of dystrophin by promoting exon skipping are approved by the FDA for the treatment of Duchenne muscular dystrophy and are effective in minimizing symptoms of the disease [48]. 

ASOs also can promote the inclusion of exons by binding and blocking regions of DNA that contain intronic splicing silencers (ISSs) or exonic splicing silencers (ESSs). Once blocked by ASOs, these regions can no longer recruit splicing repressors, resulting in improved recognition of the target exon by the spliceosome. ASO-mediated exon inclusion is a promising approach for the treatment of numerous genetic disorders. For example, the most frequent variant (c. -32-13T>GP) in the GAA gene, which causes Pompe disease, results in exon 2 skipping [49]. ASOs have been effective in promoting exon 2 inclusion in patient-derived fibroblasts possessing the splice-site variant, leading to a significant improvement in GAA activity [50]. 

The numerous mechanisms that ASOs can utilize to modulate RNA expression or splicing events give them the potential to be effective in the treatment of a vast number of rare genetic diseases. There are currently 15 ASO therapies that have been approved by the FDA for the treatment of disease, with additional clinical trials currently underway [51]. These ASOs have a range of modifications to their chemical structure and/or utilize carriers to improve their delivery to target cells. 

### 3.1. Chemical Modifications to Improve the Stability, Safety, and Delivery of Antisense Oligonucleotides 

Oligonucleotides in their natural form lack the ability to permeate cell membranes, are highly susceptible to degradation by nucleases, and only possess suboptimal binding to target sequences [52,53]. To overcome these challenges, modifications to the chemical structure of ASOs have been effective in improving ASO stability and uptake into cells. These chemical modifications also influence the ASOs’ mechanism of action once they are bound to their target. The progress in these ASO modifications over the last two decades has allowed these oligonucleotides to be translated into a clinical setting to treat patients. FDA-approved ASOs feature a variety of chemical modifications chosen to align with the drug’s intended mode of action [54].

Replacing the nonbridging oxygens on the ASOs’ phosphate group with a sulfur group, the first chemical modification of ASOs studied, results in a phosphorothioate-modified (PS) ASO [55]. The chemical structure and negative backbone charge of PS ASOs allow them to have improved protection against nuclease degradation and a strong affinity for serum proteins that significantly improves their half-lives when compared to unmodified ASOs [56,57]. 

More recently, modifications to the sugar moiety of oligonucleotides have been a promising approach to improve ASO efficacy. 2′-*O*-methyl (2′-OME) and 2′-*O*-methoxyethyl (2′-MOE) ASOs are generated by using natural ribonucleotides possessing ME or MOE groups, respectively, at the 2′ position [58]. Locked nucleic acid (LNA) is formed by creating a bridge between the 2′ oxygen and 4′ carbon on the ribose moiety, resulting in reduced conformational flexibility, which increases binding affinity [59]. These ASOs with modified sugar moieties have an improved binding affinity for the target sequence when compared to PS ASOs, and 2′-OME and 2′-MOE modifications are associated with improved safety due to reduced immune stimulation [60]. However, unlike PS ASOs, RNase H-mediated degradation is not facilitated when these modifications are used for the entirety of the ASO, so they are only suitable when steric blockage is the desired effect. However, ASOs can be designed to have a chemically unmodified centre with chemically modified nucleotides flanking to induce RNase-mediated degradation, with improved safety and specificity attributed to these modifications [61]. 

Phosphorodiamidate morpholino oligomers (PMOs), sometimes referred to as morpholino oligos, possess a backbone consisting of morpholine rings attached by phosphorodiamidate linkages. PMOs are neutrally charged, resulting in a reduced capacity to interact with proteins when bound to their target sequence [62]. Therefore, PMO-RNA binding does not promote RNase H-mediated degradation, and PMOs are used to sterically block or modulate splicing of target sequences. PMOs appear to be highly resistant to biological nucleases and enzymes and have improved aqueous solubility [60]. Due to their neutral charge, they are also well suited for conjugation with peptides, which is discussed further in this paper.

### 3.2. Antisense Oligonucleotides for the Treatment of SMA

ASOs have emerged as a powerful approach for the treatment of SMA. The severity of SMA is largely associated with *SMN2* copy number and its ability to compensate for the loss of functional SMN proteins translated from *SMN1*. ASOs have demonstrated remarkable efficiency in facilitating the inclusion of exon 7 in *SMN2* transcripts and increasing the levels of full-length SMN protein [63,64]. An ISS within intron 7, known as ISS-N1, is currently the most potent target for ASOs promoting exon 7 inclusion in *SMN2* transcripts [65]. 2′-OME ASOs, 2′-MOE ASOs, and PMOs designed to be complementary to ISS-N1 have been observed to significantly ameliorate SMA symptoms and improve the survival of SMA mice [66,67,68,69,70,71]. The delayed onset and progression of SMA symptoms through ASO treatment appears to be dose-dependent, exhibiting further improvement when administered at an earlier stage of the mouse’s life.

Nusinersen, a 2′-OME ASO targeting ISS-N1 for the treatment of SMA, was introduced to clinical trials led by Ionis Pharmaceuticals (Figure 2). Open-label phase 1 clinical trials involved intrathecal administration of Nusinersen to patients diagnosed with type 2 or type 3 SMA [72]. No safety or tolerability concerns from Nusinersen treatment were reported in phase 1 trials, and the observed increase in full-length *SMN2* expression was associated with improved Hammersmith Functional Motor Score Expanded (HFMSE) scores, indicating improved motor function, 9–14 months after administration [72]. The phase 2 open-label, dose-escalation clinical trials included patients with type 1 SMA and evaluated the tolerability of multiple doses [73]. Uptake of the ASO into motor neurons throughout the spinal cord, resulting in increased full-length SMN protein, was confirmed in treated patients and was associated with promising clinical efficiency including improved motor function and survival [73]. The observation of statistically significant improvements in motor function of patients receiving Nusinersen in placebo-controlled double-blinded phase 3 clinical trials led to the trial’s premature termination [3]. Within months of the phase 3 clinical trial ending, Nusinersen became the first FDA-approved ASO for the treatment of SMA in December 2016 [2]. ASOs alone do not possess the capability to effectively cross the blood–brain barrier (BBB), so Nusinersen is administered intrathecally, typically via lumbar puncture, to improve the delivery of these ASOs to their primary target, motor neurons [74]. 

### 3.3. Limitations of Antisense Oligonucleotides for the Treatment of SMA

Nusinersen treatment, although effective at improving symptoms of SMA in patients, is also associated with several adverse events. The most reported adverse events of Nusinersen administration in patients include lower and upper respiratory tract infections and constipation [75]. Patients receiving Nusinersen have also been observed to have a higher risk for paradoxical breathing, pneumonia, and other symptoms of respiratory dysfunction [75]. However, these symptoms are largely associated with SMA progression and are unable to be attributed specifically to Nusinersen treatment. In addition, coagulation abnormalities, thrombocytopenia, and renal toxicities have been associated with ASO treatments for different diseases [76,77]. The associated risks of bleeding complications and renal toxicity require that all patients must routinely have platelet counts and urine protein levels measured prior to treatment. 

Although chemical modifications can improve plasma stability and protect against nuclease degradation, one of the primary obstacles to improving ASO’s efficiency at eliciting their intended effect at target cells is their poor intracellular delivery [4]. ASOs, when delivered on their own, typically enter cells through endocytosis, which involves the ASO being engulfed by the cell membrane, which subsequently buds off within the cell as an ASO-containing endosome [78]. To influence their target RNA, ASOs must escape from the endosome for subsequent localization into the nucleus. If ASOs are not released from their endosome, the endosome can be fused to lysosomes, resulting in ASO degradation [79]. Poor endosomal escape is a major limiting factor of ASO efficiency [80]. Delivery systems to improve the cellular internalization of ASOs for the treatment of SMA have the potential to further improve full-length SMN levels in target cells for a superior rescue of SMA symptoms in patients at lower doses, associated with reduced side effects [81]. 

Side effects of current Nusinersen treatment are attributable not only to the ASO molecule itself but also to the invasive injection required for the drug’s delivery to the CNS. Intrathecal administration of Nusinersen is associated with post lumbar puncture side effects, including headaches and back pain [72]. The invasive nature of the procedure requires it to be conducted at a hospital or specialized facility. The frequency of required doses and the cost and burden associated with traveling to these facilities have led to many patients stopping treatment or failing to adhere to the recommended dosing schedule [82]. Developing a method to deliver ASOs that promote *SMN2* exon 7 inclusion across the BBB would facilitate the use of substantially less invasive routes of administering these therapies to patients.

## 4. Cell-Penetrating Peptides 

Some proteins have the distinct capacity to transverse cell membranes, a process known as protein transduction, without altering the membrane structure. The trans-activator of transcription (TAT) protein, belonging to the human immunodeficiency virus (HIV), was the first protein reported, in 1988, to possess this function [83,84]. The TAT protein contains a basic peptide domain that was determined to be responsible for facilitating the entry of the full-length TAT protein into cells [85]. This sequence was the first amino acid sequence to be reported as a CPP. CPPs, also known as protein transduction domains, are short (<40 amino acids) peptides that have a potent ability to penetrate biological membranes [86]. 

Since their initial discovery over three decades ago, a vast range of naturally occurring and artificially made CPPs have been reported. Due to their strong membrane-penetrating abilities, these peptides have emerged as an encouraging delivery system for drugs that would otherwise face challenges crossing these membranes. The utilization of CPPs as carriers has allowed for a significant improvement in the cellular uptake of various types of cargo, including small-molecule drugs [87], proteins [88], liposomes [89], and oligonucleotides [90]. Compared to alternative small-drug carriers currently being studied, CPPs appear to be superior due to their improved selectivity and specificity and their ability to be easily designed and synthesized [91,92]. Although recent studies have shown CPPs to be a powerful tool for small-molecule delivery, no CPPs for the delivery of therapeutics have been FDA-approved due to concerns regarding their toxicity and immunogenicity [93]. Advancing our understanding of the processes involved in their uptake and persistently assessing modifications to enhance their delivery and safety profile will facilitate the identification of superior CPP candidates appropriate for clinical translation in disease treatment. 

### 4.1. Types of Cell-Penetrating Peptides 

Over 1850 unique CPPs have been found, ranging in size, charge, solubility, and hydrophobicity [94]. Although all these features influence each CPP’s abilities, the physicochemical properties appear to be significantly regulated by the CPP’s charge [95]. As such, CPPs can be classified into three types: cationic CPPs, amphipathic CPPs, and hydrophobic CPPs. 

The majority of naturally occurring CPPs fall under the classification of cationic CPPs, including the well-researched pTAT and penetratin peptides. These CPPs are typically enriched with basic amino acid residues throughout their sequence, namely, lysine and arginine, that result in a highly positively charged peptide under normal physiological conditions [96]. The positive charge of these CPPs allows them to strongly bind to negatively charged cell membrane glycoproteins for subsequent internalization into cells [93]. Stretches of polyarginine within the peptide sequence are associated with improved cellular uptake. The optimal length for repeating arginine amino acids to improve delivery efficiency appears to be between 8 and 10 arginine residues [97]. Nuclear localization sequences (NLSs) also represent a distinct group of short-cationic CPPs that can promote nuclear import of its cargo [98]. 

Hydrophobic CPPs are composed primarily of nonpolar amino acid residues, sometimes also containing a minimal number of charged residues. Very few hydrophobic CPPs have been reported in comparison to the other types of CPPs. Those reported include C105Y [99], Pep-7 [100], and SG3 [101]. Hydrophobic CPPs have received the least attention, and their mechanism of translocation and potential applications are not well understood [102].

Over 40% of the CPPs currently reported are amphipathic CPPs, possessing both hydrophobic and hydrophilic regions [93]. Amphipathic CPPs have been derived from both naturally occurring proteins, including pVEC [103], ARF [104], and BPrPp [105], and synthetically made proteins. Synthetic amphipathic CPPs are chimeric peptides composed of a hydrophilic domain attached through a linker to an NLS [106]. Pep-1 and MPG utilize the SV40 NLS linked to HIV glycoprotein 41’s fusion protein or a peptide cluster enriched with tryptophan, respectively [106]. Many amphipathic CPPs can possess either an α-helix or β-sheet motif, associated with improved cellular uptake [93,107].

### 4.2. Internalization Mechanisms of Cell-Penetrating Peptides 

Despite the recent rigorous study of CPPs as delivery systems for small cargo/drugs, the mechanisms that allow CPPs to transverse across biological membranes are not clearly understood. CPPs and their cargo are believed to be internalized into cells through either energy-dependent endocytosis pathways or energy-independent direct-translocation pathways (Figure 3). The physicochemical properties appear to have a major effect on which pathway mediates CPP–cargo internalization; however, the experimental design, including drug concentration, cell type, and length of incubation, can also have an influence [108]. The internalization of low doses of cationic CPPs are observed to be primarily mediated by energy-independent pathways, whereas the cellular uptake of amphipathic CPPs at low doses is mediated by energy-independent pathways [86]. However, delivery of cationic CPPs through cell membranes can become mediated by direct translocation when high doses of these CPPs are used.

The cellular internalization of CPPs, particularly those delivered at low doses with large molecular weights or attached to cargo, is generally mediated by endocytosis. There are four endocytic mechanisms that have been reported to be involved in the uptake of CPPs, including macropinocytosis, clathrin-mediated endocytosis (CME), caveolae-mediated endocytosis (CvME), and clathrin- and caveolae-independent endocytosis. Macropinocytosis is the most reported endocytic pathway for CPP uptake and involves the plasma membrane deforming and protruding to encapsulate the CPP and its cargo in an endocytic vesicle that subsequently buds off the membrane and can release its contents within the cell [109]. Although micropinocytosis occurs independent of receptors, the protrusion of the cell membrane is regulated by a group of kinases and ATPases [86]. Receptor-mediated endocytosis involves the inward folding of the cellular membrane upon the CPP with a cell membrane receptor. In CME and CvME, the CPP-containing pit being formed is coated in clathrin or caveolin proteins for vesicle formation [110,111]. Receptor-mediated endocytosis can also occur independently of clathrin or caveolin; however, this process seems to occur only in specialized cells [112].

CPPs have also been observed to be internalized by cells through single-step, energy-independent processes. Different mechanisms for this process, including inverted micelle creation, pore formation, the carpetlike model, and membrane thinning, have been postulated as the mechanisms allowing for CPP direct translocation. In general, all mechanisms involve the CPP interacting with elements of the cell membrane that result in membrane instability [113]. In the presence of CPPs possessing hydrophobic regions that interact with the cell membrane, the formation of inverted micelles can occur, encapsulating the CPP and facilitating its intracellular transport [114]. The internalization of amphipathic CPPs with α-helical motifs appears to occur primarily due to pore formation. The bundles formed by the α-helical structure of the peptide interacting with the cell membrane create a gap in the membrane for their delivery inside [115]. The carpetlike model and membrane-thinning model are currently believed to be the mechanisms that allow cationic CPPs to translocate directly into cells at high doses. The carpet model involves the positively charged regions of the CPP lying parallel to the cell membrane, in a carpetlike manner, and the hydrophilic regions of the CPP orienting towards the hydrophilic regions of the membrane [109]. This interaction is thought to induce structural reorganization of the membrane, allowing the CPP to be internalized. The alternative membrane-thinning model involves the carpetlike formation of the CPP along the membrane. However, this model suggests that the interaction between the positively charged CPP and negatively charged cell membrane causes a rearrangement in the negative charges along the membrane and subsequent membrane thinning for the CPP to translocate through [116]. 

### 4.3. Cell-Penetrating Peptide Conjugation with Antisense Oligonucleotides 

As discussed earlier, ASOs face difficulties penetrating cell membranes. CPPs, with the potential to improve the cellular uptake of their cargo, can be directly conjugated to neutrally charged ASOs through maleimide, disulfide, or amide linkages. Charged molecules are suboptimal for conjugation with CPPs that also possess a charged structure due to the potential for electrostatic interactions resulting in aggregation [112]. Therefore, neutrally charged PMOs represent a commonly used backbone chemistry that is well suited for CPP conjugation. Comparing the thermal stability of PMO/target RNA heteroduplexes in both CPP-conjugated and unconjugated forms has demonstrated that CPP conjugation does not disrupt the PMO’s affinity for its target RNA and can even lead to an increase in the heteroduplex’s melting temperature [117,118].

CPPs have been assessed for their potential to enhance PMO uptake for the treatment of a spectrum of conditions, including Duchenne muscular dystrophy (DMD) [119], myotonic dystrophy type I [120], Huntington’s disease, amyotrophic lateral sclerosis [121], and SMA [5]. For example, Eteplirsen, the first FDA-approved PMO, promotes exon skipping in the dystrophin gene to restore disrupted reading frames [122]. However, high renal clearance of the drug in patients is apparent, and high doses of the drug are required for clinical efficiency [123]. Multiple CPPs, including Pip5e, Pip6a, M12, and DG9, conjugated to PMOs mediating exon skipping for dystrophin reading frame restoration have resulted in improved cellular uptake of the PMO as well as increased levels of dystrophin protein [112,124,125,126]. 

## 5. Cell-Penetrating Peptide Conjugated Antisense Oligonucleotides for the Treatment of Spinal Muscular Atrophy 

Given the increasing evidence exhibiting CPPs’ notable enhancement of ASO uptake in target cells, several studies have been undertaken employing CPPs to deliver exon inclusion-mediated ASOs for the treatment of SMA (Table 1). Nusinersen consists of 2′-OME modifications that result in an anionic molecule [127]. Since the chemical structure of Nusinersen is not ideal for CPP conjugation, studies exploring CPP-ASO conjugates for SMA treatment have opted to use PMOs targeting the same or a similar region of *SMN2* pre-mRNA as Nusinersen.

In 2016, Hammond et al. developed and evaluated a PMO-internalizing peptide (Pip), known as Pip6a, conjugated to a PMO targeting *SMN2*’s exon 7 ISS-N1 element [5]. Either Pip6a-PMO conjugates or PMO alone were administered to a severe mouse model of SMA through facial vein injection at PD-0 [128]. The mice treated with unconjugated PMOs displayed no survival improvements compared to saline-treated mice, which survived for a median of 12 days [5]. In contrast, Pip6a-PMO treatment extended median survival to 167 days. The Pip6a-PMOs not only significantly prolonged survival but demonstrated observable improvements in weight, muscle strength, coordination, and functional neuromuscular junctions [5]. Furthermore, the analysis of CNS tissues from *SMN2* transgenic adult mice treated intravenously with Pip6a-PMO conjugates confirmed the capability of CPPs to deliver PMOs to the CNS, thereby elevating full-length SMN levels in target cells [5]. 

Shabanpoor et al. synthesized and assessed various CPP sequences sourced from naturally occurring proteins, such as RVG, Angiopep-2, ApoE, THR, and TET-1, conjugated to a 20-mer PMO known for promoting exon 7 inclusion in *SMN2* [6,129]. These CPPs, including RVG, Angiopep2, THR, Pepc7, SynB3PFV, and a branched derivative of ApoE termed Br-ApoE (K->A), were administered to SMA type 1 patient fibroblasts to determine their efficacy in improving full-length SMN expression levels. All CPP-PMO conjugates demonstrated a dose-dependent increase in SMN expression, with Br-ApoE (K->A) exhibiting the highest efficiency [6]. Neonatal SMA mice received treatment with CPP-PMO conjugates, with unconjugated PMOs serving as a control, via facial vein injection at PD-4. Although increased levels of full-length SMN were observed in the brain, kidney, and quad of CPP-PMO-treated mice, statistically significant differences were evident only with Br-ApoE (K->A) [6]. 

A challenge associated with ASO-CPP delivery to the CNS is that they appear to become entrapped in, and subsequently cleared from, the endo-lysosomal barrier [130]. Therefore, Dastpeyman et al. evaluated the efficiency of a variety of CPPs with the novel capacity for endosomal escape [7]. Initial studies compared ApoE (141–150) and ApoE (133–150), which utilize receptor-mediated endocytosis for uptake, with CLIP6 and pVEC, which both appear to use-endosomal pathways for cellular uptake, as conjugates for PMOs that promote *SMN2* exon 7 inclusion. All four peptides improved the delivery of PMOs in SMA patient fibroblasts, but ApoE (133–150) conjugated to PMOs showed a far superior improvement in full-length SMN levels in comparison to the other CPPs [7]. Chimeric CPPs composed of ApoE (141–150) linked at its N-terminal domain with either HA2, GALA, or NLS peptides, which possess endosomal-disrupting capabilities, or four histidine residues (H4) were observed to further improve PMO uptake [7]. The strongest CPP, ApoE (133–150), linked to the strongest endosomal escape peptide, HA2, was conjugated to PMOs and administered to *SMN2*^+/−^ transgenic mice through tail vein injection, resulting in a significant improvement in SMN expression in the brain and spinal cord [7].

In 2022, Bersani et al. evaluated the delivery of a previously validated PMO targeting the ISS-N1 region of *SMN2*, known as HSMN2Ex7D (10–34) [131], conjugated to four CPPs: TAT, R6, r6, and (RXRRBR)_2_XB, also known as RXR [8]. Administered intracerebroventricularly (ICV) or intravenously (IV) to heterozygous SMAΔ7 mice, the PMO-CPP conjugates exhibited markedly elevated levels of full-length SMN protein in the brain and spinal cord compared to HSMN2Ex7D delivered alone. r6 and RXR, both arginine-rich CPPs, showed superiority over the other two CPPs in initial experiments. Therefore, RXR-PMO and r6-PMO conjugates intraperitoneally (IP) injected into symptomatic SMAΔ7 mice at PD-5 were observed to be taken up well in the CNS. The increased presence of full-length SMN in the CNS resulted in a significant improvement in neuromuscular connections and improved survival, extending to 23 and 42 days for RXR-PMO and r6-PMO, respectively [8].

In 2023, Aslesh et al. investigated the potential of the DG9 peptide, derived from the human polyhomeotic 1 homolog transcription factor, as a PMO conjugate in SMA mice [9]. Conjugated to an 18-mer PMO with the same sequence as Nusinersen, the DG9 CPP was subcutaneously administered to SMA Taiwanese mice at PD-0. Compared to R6G-PMOs, used as a benchmark control peptide PMO conjugate, as well as unconjugated PMOs and 2′-MOEs, DG9-PMO treatment yielded the most improved muscle strength and improved median lifespan, increasing it from 12 days for unconjugated PMOs to 58 days for DG9-PMO-treated mice [9]. To assess DG9-PMO uptake into the CNS at a more advanced age, a milder SMA mouse model received subcutaneous injections of DG9-PMO at PD-5. The improved CNS uptake of DG9-PMOs in comparison to unconjugated ASOs further underscores DG9 as a promising delivery system for enhancing ASO efficacy in SMA treatment. 

## 6. Conclusions

The compelling outcomes demonstrated by the studies discussed above underscore the potential of CPPs as a highly promising strategy to enhance ASO delivery for SMA treatment. However, their translation to the clinic is impeded by the limited understanding of their in vivo stability, immunogenicity, and cellular toxicity. Additional investigations aimed at comprehensively characterizing and addressing these limitations as well as further elucidating the mechanisms facilitating CPPs’ penetration through cell membranes and the BBB are anticipated. Such efforts are crucial for the identification of superior CPP candidates capable of safely and efficiently delivering these PMOs, thereby facilitating their clinical translation. 

## Figures and Tables

**Figure 1 molecules-29-02658-f001:**
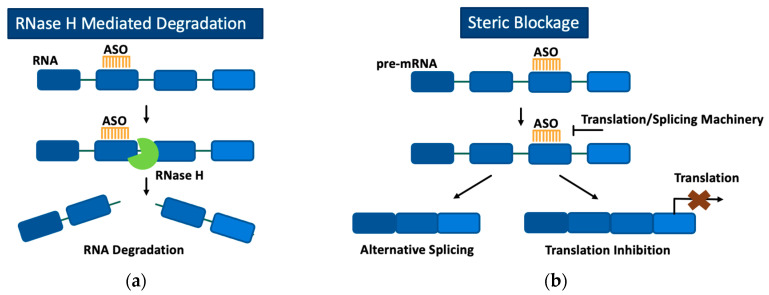
Mechanism of action of ASOs through either RNase H-mediated degradation (**a**) or steric blockage (**b**). (**a**) ASOs bind to target RNA, and the RNA-ASO complex is recognized by RNase H, which subsequently degrades the target RNA. (**b**) ASOs bind to target RNA and, based on the region targeted, block translation or splicing machinery. Blocking splicing machinery results in alternative splicing of the pre-mRNA transcript. Blocking translation machinery prevents translation of the target RNA.

**Figure 2 molecules-29-02658-f002:**
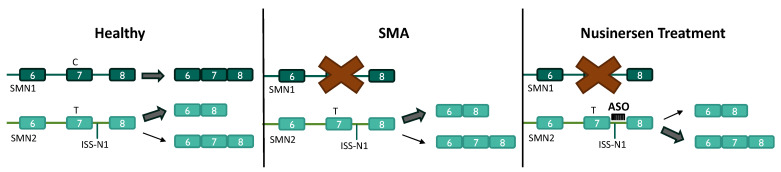
Nusinersen mechanism of action for the treatment of SMA. In healthy cells, *SMN1* is processed primarily into full-length SMN mRNA, containing exon 7. Exon 7 is predominantly skipped in *SMN2* pre-mRNA processing, resulting in a dysfunctional SMN protein. In SMA, no functional SMN protein is translated from *SMN1*. Nusinersen, an ASO targeting the ISS-N1 element, promotes the inclusion of exon 7 in *SMN2* pre-mRNA processing to increase full-length SMN proteins.

**Figure 3 molecules-29-02658-f003:**
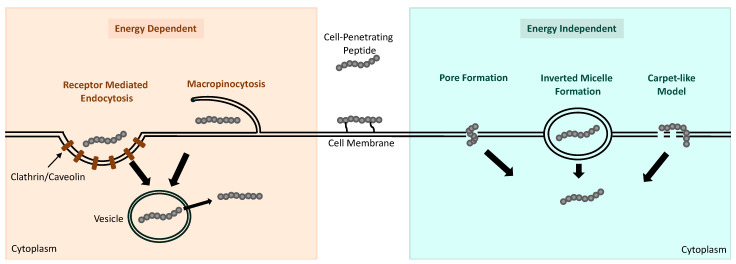
Schematic representation of the mechanisms reported to facilitate the uptake of CPPs into cells. Energy-dependent pathways include receptor-mediated endocytosis and micropinocytosis, which result in the CPP being internalized in an endocytic vesicle for subsequent release into the cytoplasm. Energy-independent mechanisms include pore formation, inverted micelle formation, and the carpetlike model, which allow the CPP to enter the cell directly.

**Table 1 molecules-29-02658-t001:** Notable cell-penetrating peptides that are found to improve PMO efficiency and uptake through the CNS in SMA mice.

Cell-Penetrating Peptide (CPP)	CPP Sequence	Route of Administration	Reference
Pip6a	RXRRBRRXRYQFLIRXRBRXRB	IV injection	[5]
Br-ApoE (K->A)	LRALRARLLR-G*GKX-Bpg-G (LRALRARLLR-G*G)	IV injection	[6]
HA2-ApoE (133–150)	GLFHAIAHFIHGGWH-X-LRVRLASHLRKLRKRLLR	IV injection	[7]
RXR	[RXRRBR]2XB	IP and ICV injection	[8]
r6	RRRRRR	IP and ICV injection	[8]
DG9	YArVRRrGPRGYArVRRrGPRr	SQ injection	[9]

(*) indicates the branch point of the peptide. X, 6-aminohexanoic acid; B, β-alanine spacer; Bpg, bishomopropargylglycine; lowercase r, d-arginine; IV, intravenous; IP, intraperitoneal; ICV, intracerebroventricular; SQ, subcutaneous.

## Data Availability

No new data were created or analyzed in this study. Data sharing is not applicable to this article.

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
