# Peer review of "Potential of Cell-Penetrating Peptide-Conjugated Antisense Oligonucleotides for the Treatment of SMA"

_molecules, 2024, doi:10.3390/molecules29112658_

Round 1

Reviewer 1 Report

Comments and Suggestions for Authors

The manuscript reviews the development of therapeutic oligonucleotides aimed to treat spinal muscular atrophy (SMA) focusing in recent research on peptide-oligonucleotide conjugates carrying cell-penetrating peptides. These conjugates are specially interesting with oligonucleotides carrying neutral internucleotide linkages such as PMO oligonucleotides that are especially active in the modulation of RNA splicing mechanisms that is the main mechanism of action of therapeutic oligonucleotides aimed to treat SMA. The review is written with a good combination of basic introductory text together with biomedical interest and the impact of the latest developments in this emerging field.  The text is well written and maintains the interest along the manuscript. In my opinion the manuscript can be accepted after very minor changes.

  1. The address of the authors lacks the country (Canada).

  2. Page 4, line 165, it says “…results in a phosphorothioates (PS) ASO.” I believe it should be  “…results in phosphorothioate (PS)-modified ASOs.”

  3. Page 4, line172, it says “—by including an oxygenated group on the 2’ position”. This is not true as these derivatives are not made by including an oxygenated group on the 2’-position but, on the contrary using the natural ribose nucleoside that contains an OH in 2’-position. I believe it should say “by using the natural ribonucleotides functionalized with ME and MOE groups at the 2’-OH position.”

  4. Table 1. The abbreviations X, Y, B, Bpg, Ar and Rr in the CPP sequences are not defined. Most probably indicate non-natural amino acids.

Author Response

RESPONSE TO REVIEWER ONE:

The manuscript reviews the development of therapeutic oligonucleotides aimed to treat spinal muscular atrophy (SMA) focusing in recent research on peptide-oligonucleotide conjugates carrying cell-penetrating peptides. These conjugates are specially interesting with oligonucleotides carrying neutral internucleotide linkages such as PMO oligonucleotides that are especially active in the modulation of RNA splicing mechanisms that is the main mechanism of action of therapeutic oligonucleotides aimed to treat SMA. The review is written with a good combination of basic introductory text together with biomedical interest and the impact of the latest developments in this emerging field.  The text is well written and maintains the interest along the manuscript. In my opinion the manuscript can be accepted after very minor changes.

1. The address of the authors lacks the country (Canada).

Response: This comment has been addressed and the country (Canada) has been included in the authors’ addresses. 

2. Page 4, line 165, it says “…results in a phosphorothioates (PS) ASO.” I believe it should be “…results in phosphorothioate (PS)-modified ASOs.”

Response: We appreciate the suggestion to change “phosphorothioates (PS) ASO” to phosphorothioate (PS)-modified ASOs. In current literature, these ASOs are normally referred to as PS ASOs. Therefore, line 165 has been updated to “phosphorothioate-modified (PS) ASO.”

3. Page 4, line172, it says “—by including an oxygenated group on the 2’ position”. This is not true as these derivatives are not made by including an oxygenated group on the 2’-position but, on the contrary using the natural ribose nucleoside that contains an OH in 2’-position. I believe it should say “by using the natural ribonucleotides functionalized with ME and MOE groups at the 2’-OH position.”

Response: Thank you for the valuable suggestion to update line 172 to better reflect how these ASOs are modified. Line 172 has been updated from “by including an oxygenated group on the 2’ position” to “using natural ribonucleotides possessing ME or MOE groups, respectively, at the 2’ position.”

4. Table 1. The abbreviations X, Y, B, Bpg, Ar and Rr in the CPP sequences are not defined. Most probably indicate non-natural amino acids.

Response: We appreciate your comment acknowledging the absence of the defined abbreviations of non-natural amino acids in Table 1. The abbreviations for X, B, Bpg, and lowercase r have 

Reviewer 2 Report

Comments and Suggestions for Authors

The review provides a comprensive analysis of ASO therapy, their application for the treatment of SMA, and the encouraging potential of CPPs as delivery systems to improve ASO uptake and overall efficiency.

1) In lane 15 the authors should specify that both SMN2 and SMN1 are genes.

2) In lane 96 the authors wrote: “Short fragments of nucleic acid analogs, known as nucleotides, have become an incredibly useful tool for understanding gene function and represent a promising avenue for genetic disease therapeutics. 

3) In lane 97 the authors should remove “analogs” because they refer to standard oligonucleotides. The authors describe in the paragraph 3.1 the nucleic acid analogs.

4) CPPs have been assessed for their ability to promote PMO uptake. Authors should explain, using references also, why the conjugation of a CPP (<40 amino acids) to the PMO does not affect the stability (CD or UV melting techniques) of the interaction between the PMO and its complementary target sequence.

Comments on the Quality of English Language

The manuscript is well written, well described with a good English language. 

Author Response

RESPONSE TO REVIEWER TWO:

The review provides a comprensive analysis of ASO therapy, their application for the treatment of SMA, and the encouraging potential of CPPs as delivery systems to improve ASO uptake and overall efficiency.

1. In lane 15 the authors should specify that both SMN2 and SMN1 are genes.

Response: We appreciate your valuable suggestion. Lane 15 has been changed from “It can effectively promote alternative splicing in SMN2, an analog of SMN1…” to “It effectively promotes alternative splicing in pre-mRNA transcribed from the SMN2 gene, an analog of the SMN1 gene.”

2. In lane 96 the authors wrote: “Short fragments of nucleic acid analogs, known as nucleotides, have become an incredibly useful tool for understanding gene function and represent a promising avenue for genetic disease therapeutics. 

Response: Comment two appears to be continued in comment 3. The response to both comments can be seen below in the response to comment 3.

3. In lane 97 the authors should remove “analogs” because they refer to standard oligonucleotides. The authors describe in the paragraph 3.1 the nucleic acid analogs.

Response: We appreciate your concern regarding the term “analog” in lane 97 of our manuscript. The concern has been addressed and “analogs” on lane 97 has been updated to “polymers”, a more appropriate term for the description of oligonucleotides.

4. CPPs have been assessed for their ability to promote PMO uptake. Authors should explain, using references also, why the conjugation of a CPP (<40 amino acids) to the PMO does not affect the stability (CD or UV melting techniques) of the interaction between the PMO and its complementary target sequence.

Response: We sincerely appreciate your valuable input. To address this comment, we have included an explanation of the thermal stability of PMO/RNA heteroduplexes when conjugated to CPPs in lines 385 to 389, which reads: “Comparing the thermal stability of PMO/target RNA heteroduplexes in both CPP-conjugated and unconjugated forms has demonstrated that CPP conjugation does not disrupt the PMO’s affinity for its target RNA, and can even lead to an increase in the heteroduplex’s melting temperature”. References from studies utilizing CD and PCR melting techniques to evaluate the CPP-PMOs affinity for target sequences have been included.